# Association between Epstein-Barr virus reactivation and severe malaria in pregnant women living in a malaria-endemic region of Cameroon

**Ide Armelle Djuidje Chatue**[1,2,3], **Maximilienne Ascension Nyegue**[1], **Severin Donald Kamdem**[2,4], **Franklin Maloba**[2,5,6], **Iqbal Taliy Junaid**[3], **Pawan Malhotra**[3]*, **Palmer Masumbe Netongo**[2,6,7]*

1 Department of Microbiology, University of Yaounde I, Yaounde, Centre, Cameroon, 2 Molecular Diagnostics Research Group, Biotechnology Centre-University of Yaounde I (BTC-UYI), Yaounde, Centre, Cameroon, 3 Malaria Biology, International Centre for Genetic Engineering and Biotechnology (ICGEB), New Delhi, Delhi, India, 4 Department of Pathology, University of Utah School of Medicine, Salt Lake City, Utah, United States of America, 5 Department of Microbiology and Parasitology, University of Buea, Buea, Southwest Cameroon, 6 Biology Program, School of Science, Navajo Technical University, Crownpoint, New Mexico, United States of America, 7 Department of Biochemistry, University of Yaounde I, Yaounde, Centre, Cameroon

* masumben@gmail.com (PMN); pawanmal@gmail.com (PM)

**Data Availability Statement:** All relevant data are within the manuscript and its Supporting Information files.

## Abstract

Malaria kills nearly 619,000 people each year. Despite the natural immunity acquired to malaria, pregnant women and children under five die from severe forms of the disease in sub-Saharan Africa. Co-infection with acute Epstein-Barr Virus (EBV) infection has been shown to suppress the anti-malarial humoral responses, but little is known about the impact of EBV reactivation on malaria-associated morbidity. This study investigated the association between EBV reactivation and malaria severity in pregnant women living in a malaria-endemic region in Cameroon. A cross-sectional study was conducted on 220 pregnant women attending antenatal consultations in three health facilities in the West region of Cameroon. Malaria was diagnosed by microscopy, and *Plasmodium* species were identified by Nested PCR. Plasma samples were analyzed by ELISA for the presence of EBV nuclear antigen, EBV viral capsid antigen, and EBV early antigen to determine EBV reactivation. All statistics were performed using GraphPad Prism and SPSS software. The prevalence of malaria among pregnant women was 23.2%, of which 18.6% were *P. falciparum* mono-infections and 4.5% mixed infections (3.6% *P. falciparum* and *P. malariae*; 0.9% *P. falciparum* and *P. ovale*). 99.5% of the women were EBV seropositive, and 13.2% had EBV reactivation. Pregnant women with reactivated EBV were more likely to develop severe malaria than pregnant women with latent EBV (OR 4.33, 95% CI 1.08–17.25, p = 0.03). The median parasitemia in pregnant women with latent EBV was lower than in those with EBV reactivation (2816 vs. 19002 parasites/μL, p = 0.02). Our study revealed that lytic reactivation of EBV may be associated with the severity of malaria in pregnant women. Suggesting that, like acute infection, EBV reactivation should be considered a risk factor for severe malaria in

**Funding:** The first phase of this work was supported by funds from the "The African coaLition for Epidemic Research, Response and Training, (ALERRT). "ALERRT" is part of the European and Developing Clinical Trial Partnership (EDCTP2) Programme 2 supported by the European Union under Grant Agreement RIA2016E-1612 (https://www.alerrt.global/content/alerrt-african-coalition-epidemic-research-response-and-training). This second phase of the work was supported by the Italian Ministry of Foreign Affairs and International Cooperation, and the International Centre For Genetic Engineering and Biotechnology (ICGEB) (https://www.icgeb.org/) through the WE-STAR - WS/CMR22-01 mobility fellowship awarded to Ide Armelle DJUIDJE CHATUE (IADC) and by JC Bose fellowship grant (DST/20/015). PM is an Arturo Falaschi Emeritus Scientist (AFES) of ICGEB and J. C Bose Fellow of SERB Govt of India. The funders had no role in study design, data collection and analysis, decision to publish, or preparation of the manuscript.

**Competing interests:** The authors have declared that no competing interests exist.

pregnant women in malaria-endemic regions or could serve as a hallmark of malaria severity during pregnancy. Further detailed studies are needed.

## Introduction

Malaria is a deadly tropical disease caused by an obligate intracellular protozoan of the genus *Plasmodium* that infects human hosts through the bites of infected female Anopheles mosquitoes [1]. In 2021, the World Health Organization estimated that around 247 million people were infected worldwide, and 619,000 died [2] due to malaria; ~34% rise in deaths compared to pre-Covid-19 pandemic in 2019 [3]. Malaria accounts for 29.9% of consultations and 64% of hospitalizations in Cameroon. According to the Cameroon National Malaria Control Programme, around 6 million malaria cases and 4,000 deaths were recorded in 2022 [4, 5]. Malaria during pregnancy is a major public health concern for developing countries, especially in sub-Saharan Africa. In 2021, approximately 13.3 million pregnant women living in African regions were exposed to malaria infection during pregnancy, and 961,000 infants born from these women had a low birth weight [2]. The consequences of malaria during pregnancy include severe maternal anemia, low birth weight, premature delivery, and maternal and infant mortality [6–8].

In sub-Saharan Africa, where malaria is endemic, co-infections with one or more pathogens are common due to poor health and socio-economic conditions [9, 10]. One of the most common pathogens is the Epstein-Barr virus, a human herpes virus that infects more than 95% of the world's population and persists throughout life [11, 12]. The virus is transmitted mainly orally, by direct contact with contaminated saliva, or indirectly through blood transfusion, organ or tissue transplantation, sexual intercourse, and breast milk [13–15]. EBV infection in children usually doesn't cause symptoms. However, the most common primary symptoms of EBV-infected children include fever, cough, skin eruption, lymphadenopathy, eyelid edema, and pharyngalgia [16]. In adolescents and young adults, the primary infection clinically manifests as acute infectious mononucleosis [11, 17] or can cause autoimmune diseases and lymphoproliferative malignancies in immunocompromised individuals [18, 19]. During pregnancy, 35% of pregnant women show EBV reactivation due to the decrease in cellular immunity [20]. The consequences of EBV reactivation in pregnancy include severe symmetrical fetal growth retardation, low birth weight, stillbirth, congenital malformations, and a shorter duration of pregnancy [21–23].

There is compelling evidence that co-infections with the Epstein-Barr virus worsen the health conditions of people infected with *P. falciparum* in malaria-endemic areas [24–26]. Previous studies have shown that acute infection with a gammaherpesvirus 68 (MHV68), such as EBV, negatively influences the development of the humoral response to secondary *P. falciparum* infection, transforming a non-lethal infection into a lethal one [27, 28]. Recently, a study revealed that persistent EBV DNA in the peripheral blood of adults in a *P. vivax* semi-immune population alters the antibody response to major malaria vaccine candidates (DEKnull-2) [29]. It has also been suggested that the reactivation of EBV infection may facilitate the development of cerebral malaria. Indeed, Indari *et al.* demonstrated that during malaria, EBV reactivation increased red blood cells adhesion to the human brain endothelial cells and significantly elevated inflammatory markers, contributing to the exacerbation of cerebral malaria [30].

However, the impact of the reactivation of EBV on the severity of malaria during pregnancy has not been studied yet. Although higher EBV viral loads and EBV-specific antibody levels

have been observed in pregnant women with malaria than in pregnant women without malaria [31], the association between EBV reactivation and malaria severity during pregnancy remains unclear. Besides this, it is still unclear that some pregnant women in *P. falciparum*-endemic areas naturally clear the parasites [32, 33], while others are more likely to develop severe forms of the disease. This study aimed to investigate the association between EBV reactivation and severe malaria in pregnant women living in a malaria-endemic region of Cameroon and to assess if EBV reactivation increases the severity of malaria attacks in pregnant women.

## Materials and methods

### Study design

A cross-sectional study was conducted between January and September 2022 on pregnant women of all gravida attending antenatal consultations or hospitalized in the maternity or gynecology services of the different hospitals. Data were collected during the dry and rainy seasons when transmission of *Plasmodium* is high [34]. A well-structured questionnaire was administered to participants who voluntarily agreed to participate in the study to collect socio-demographic data.

### Study site

The study was conducted in two health facilities in the Nde division (Bangangte District Hospital (BDH) and Universite des Montagnes Teaching Hospital (CUM)) and one health facility in the Mifi division [Mifi District Hospital of Bafoussam (MDHB)] in the West Region of Cameroon (Fig 1). These hospitals provide health services that are accessible to the public with high-quality and low-cost antenatal care for pregnant women. Bafoussam and Bangangte are the heads of the Mifi and Nde divisions. Bangangte covers an area of 829 km$^2$ with an estimated population of 200,000, while Bafoussam covers an area of 402 km$^2$ with a population of 347,517. These two semi-urban towns are located in the West region of Cameroon (5°30′N and 10°30′E), an area made up of mountains, plains, and plateaux at altitudes of between 1,000 and 1,500 meters. The climate is humid tropical, with heavy rainfall favoring the proliferation of Anopheles mosquitoes because of a rainy season lasting around seven months (from late March to October) and a dry season lasting five months (from late October to early March) [35]. Malaria transmission is relatively stable in the region. Entomological inoculation rates vary from 62.8 to 90.5 infectious bites/per person/year [36, 37].

### Study population

The study population consisted of pregnant women of all gestational ages and gravida attending antenatal consultation or hospitalized at the three selected hospitals. Participants approached the hospital with at least one of the following symptoms: fever (temperature ≥ 38°C) accompanied by chills, asthenia, fatigue, headache, sore throat, sore muscle, sweat and thrill, pallor, jaundice, dizziness, abdominal pain, and respiratory distress. The inclusion criteria were as follows: consenting pregnant women of all gestational ages with at least one of the above symptoms and seronegative for Human Immunodeficiency Virus (HIV), Hepatitis B Virus (HBV), and Hepatitis C Virus (HCV). We excluded all pregnant women seropositive for cytomegalovirus, intestinal worms, toxoplasmosis (IgM), or any infections mentioned in the inclusion criteria; pregnant women on corticosteroids, acyclovir, antihistamines, antibiotics, and antimalarial; and those with confirmed depression because these factors may constitute a bias for the rest of this research work.

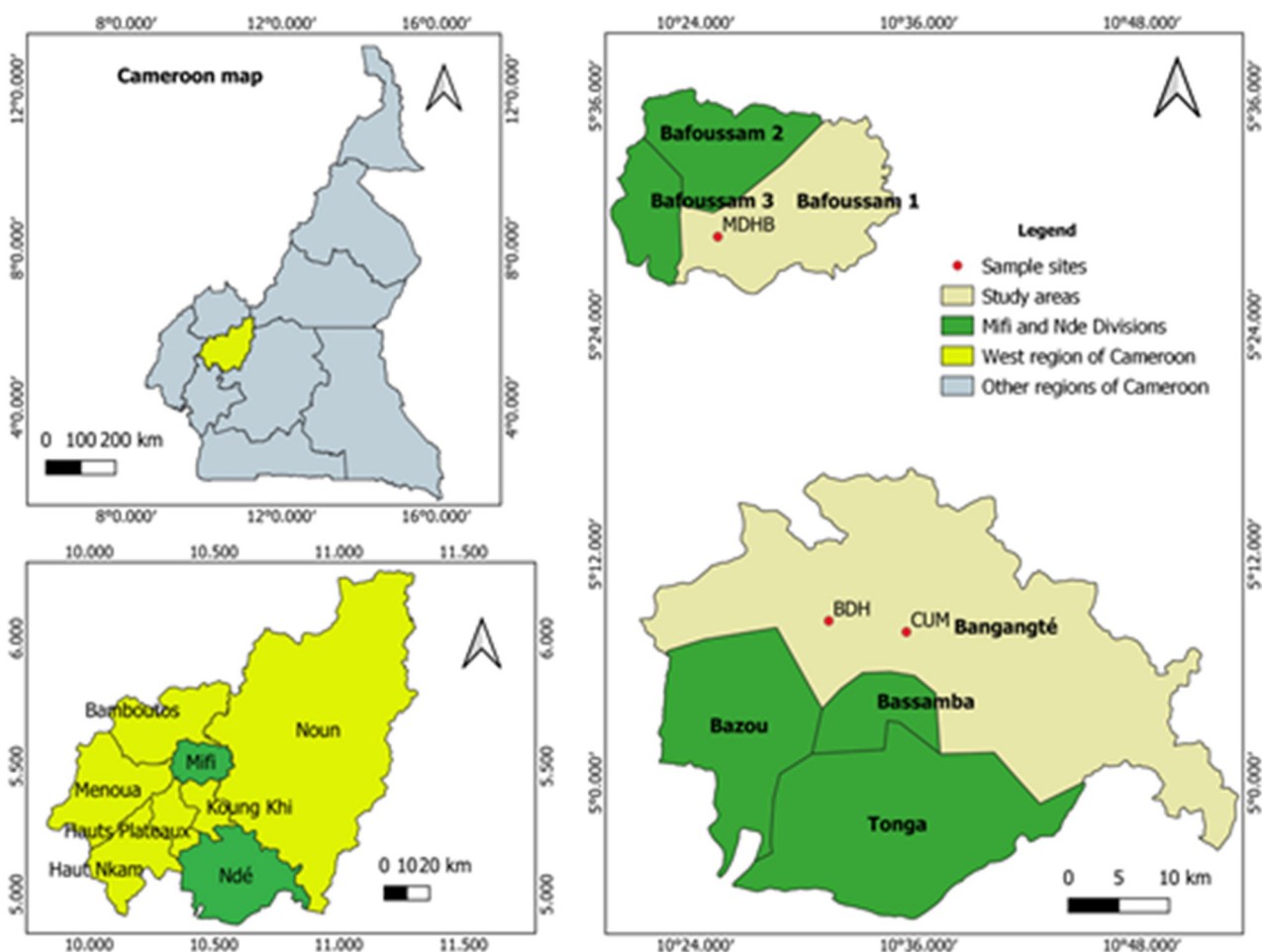

**Fig 1. Illustrative map showing the West region of Cameroon with different study sites.** BDH: Bangangte District Hospital, MDHB: Mifi District Hospital of Bafoussam; CUM: Universite des Montagnes Teaching Hospital of Bangangte. Source: Own elaboration. The map was created using the software QGIS version 3.32.3-Lima. *https://qgis.org/*. QGIS is licensed under the GNU General Public License version 2, *https://www.gnu.org/licenses*. The base layer of the Cameroon map was obtained on the website *http://www.diva-gis.org/gdata* and data were extracted from the GADM database (*www.gadm.org*), version 2.5.

## Ethical statement

This study has been approved by the National Ethics Committee for Human Health Research of Yaounde (N˚2021/12/106/CE/CNERSH/SP). Administrative authorizations were obtained from all health establishments. Written consent was obtained from each participant or their parents/guardians (for minors) before registration in the study. Positive malaria cases were referred to clinicians at the hospital for appropriate therapeutic management.

## Sample size determination

Participants were recruited by random sampling during the antennal consultation. The sample size (N) was calculated using Cochran's formula for large population proportion [38]: [$N = Z^2_{\alpha/2} P(1-P) / E^2$]. Where Z is the area under the acceptance region in a normal distribution (1 – α), it is 1.96 for a 95% confidence interval. P is the prevalence of EBV and malaria coinfection among pregnant women in Kenya, which is 64% [31]. Using a tolerated error of 5% (E), the sample size required for this study was 354 participants.

## Data collection

A well-structured questionnaire in both official languages (English and French) was used to collect socio-demographic data (sex, age, residence, weight, environment, employment status, marital status, level of education, ethnicity, and maternal income); obstetric factors (gravidity, parity, trimester of pregnancy); clinical symptoms; malaria prevention measures (use of insecticide-treated mosquito nets, Intermittent Preventive Treatment); and risk factors for EBV reactivation in participants (alcohol, stress, anxiety, illicit substances [39–42]). Insecticide-treated net use was defined as Yes for those who had or used a net and No for those who did not have or use a net. IPT use was defined as follows: Yes, for those who took at least one dose of IPT during pregnancy and No for those who did not take IPT.

## Blood sample collection

Approximately 5 mL of blood was collected from each participant by venipuncture into an ethylenediaminetetraacetic acid (EDTA) tube. All samples were transported using a medical cooler box and sent to the Laboratory of Microbiology of "Universite des Montagnes" Teaching Hospital in Bangangte for analysis. The blood collected was used to diagnose malaria and determine hemoglobin levels. The plasma extracted after centrifugation was used to diagnose EBV infection. Up to 50 μL blood spotted on Whatman 903TM filter paper (Lasec, Cape Town, South Africa) was stored at room temperature for the molecular identification of *Plasmodium* species.

## Diagnosis of malaria

**Microscopy of malaria.** A thick blood smear was prepared to detect malaria parasites as described by WHO [43]. The number of parasites was counted per 200 leukocytes, and the parasite density was calculated assuming a total WBC count of 8,000 cells/μL of blood [44]. Parasite density was classified into three categories: Low (1–999 parasites/μL), Moderate (1000–9999 parasites/μL), and High ($\geq$10000 parasites /μL) [45]. Pregnant women with high parasitemia associated with one or more of the manifestations of severe malaria, such as were classified as severe malaria cases according to WHO [46]. Pregnant women with a parasitemia of less than 10,000 parasites/μL and who did not have these characteristics were considered uncomplicated malaria cases.

**Nested PCR.** DNA was extracted from Whatman 903 TM cards using the phenol-chloroform method as described previously [47, 48]. The identification of *Plasmodium* species was performed using the nested PCR technique. Specific sequences of *P. falciparum*, *P. ovale*, *P. vivax*, and *P. malariae* 18S rRNA small subunit genes were amplified as described by Snounou *et al*. [49] with minor modification of the amplification conditions. PCR products were purified using the QIAquick PCR Purification Kit (Cat N˚28106, Hilden, Germany) and sent for sequencing to Macrogen in South Korea to confirm the species identified.

**Diagnosis of EBV infection.** Patient plasma samples were analyzed for the presence and level of viral capsid antigen immunoglobulin IgM (VCA), nuclear antigen immunoglobulin IgG (NA-1), and early antigen IgG (EA-D) using commercial ELISA kits (TestLine Clinical Diagnostics s.r.o, Czech Republic) according to the manufacturer's instructions. Quantitative results were interpreted as follows: antibody levels >22 U/ml were considered positive; antibody levels <18 U/ml were considered negative, and antibody levels between 18 to 22 were considered borderline area. For semi-quantitative results, the positivity index (PI) was calculated as follows: PI = Absorbance of the sample / mean absorbance of the cut-off. The results were interpreted based on the following index value thresholds: Negative if IP <0.9, positive if IP > 1.1, and borderline if 0.9<IP <1.1, following the manufacturer's instructions.

Epstein-Barr virus reactivation was defined by the presence of IgG EBNA-1 along with IgG EA or IgG EBNA-1 with IgM VCA or both IgG EA and IgM VCA (Table 1) [50–52].

## Statistical analysis

The data collected was entered into an Excel spreadsheet and then analyzed using GraphPad Prism 9.0 software and SPSS version 23.0. Quantitative variables were represented by the median (IQR), while frequency and percentage were used to summarise qualitative data. The Mann-Whitney U test was used for comparison between two groups, and the Kruskal-Wallis test was used for comparisons across multiple groups. Multivariate logistic regression was used to determine factors associated with malaria, and univariate logistic regression was used to assess the association between EBV reactivation and severe malaria. Simple linear regression was used to determine the correlation between parasitemia and the level of EBV antibodies. Statistically significance was considered as $p$-value <0.05 at the 95% confidence interval.

## Results

### Study flow diagram

A total of 371 symptomatic pregnant women attending antenatal clinics were interviewed in three hospitals in the West region of Cameroon. Of these, 296 voluntarily agreed to participate in the study. Serological diagnosis for HIV, HBV, HCV, and toxoplasmosis (IgM) was performed on consenting participants (Fig 2).

### Sociodemographic and obstetric characteristics of the study population

Using a well-structured questionnaire, sociodemographic data were collected and presented in Table 2. The median age of the participants was 26 years (IQR: 23–31 years), and the median gestational age was 21 weeks (IQR: 16–27 weeks). Ninety participants (40.9%) were married, 151 (68.6%) had secondary education, 146 (66.4%) lived in urban areas, and 69 (31.4%) were unemployed. In addition, 141 participants (64.1%) were multiparous, and 123 (55.9%) were in the second trimester of pregnancy. The study population was diverse, with a predominance of the Bamileke ethnic group (92.7%).

### Prevalence of malaria among pregnant women and risk factors

Of the 220 pregnant women, 51 (23.2%) had symptomatic malaria with positive microscopy and PCR. Multivariable logistic regression analysis of associated factors for malaria is presented in Table 3. Pregnant women living in urban areas were less likely to be infected with *Plasmodium* compared with those living in rural areas (AOR 0.12, 95% CI 0.07–0.27). The use of Long-lasting insecticidal nets (LLINs) and Intermittent Preventive Treatment (IPT) had a protective effect against malaria (AOR 0.35, 95% CI 0.14–0.84, and AOR 0.23, 95% CI 0.09–

**Table 1. Clinical status of Epstein-Barr infection according to viral serology results.**

| Clinical Status | VCA IgM | EA IgG | EBNA-1 IgG |
|---|---|---|---|
| No previous infection | - | - | - |
| Acute infection | + | + | - |
| Past infection | - | - | + |
| Reactivation | +/- | + | +/- |

(-) antibody absence; (+) antibody presence

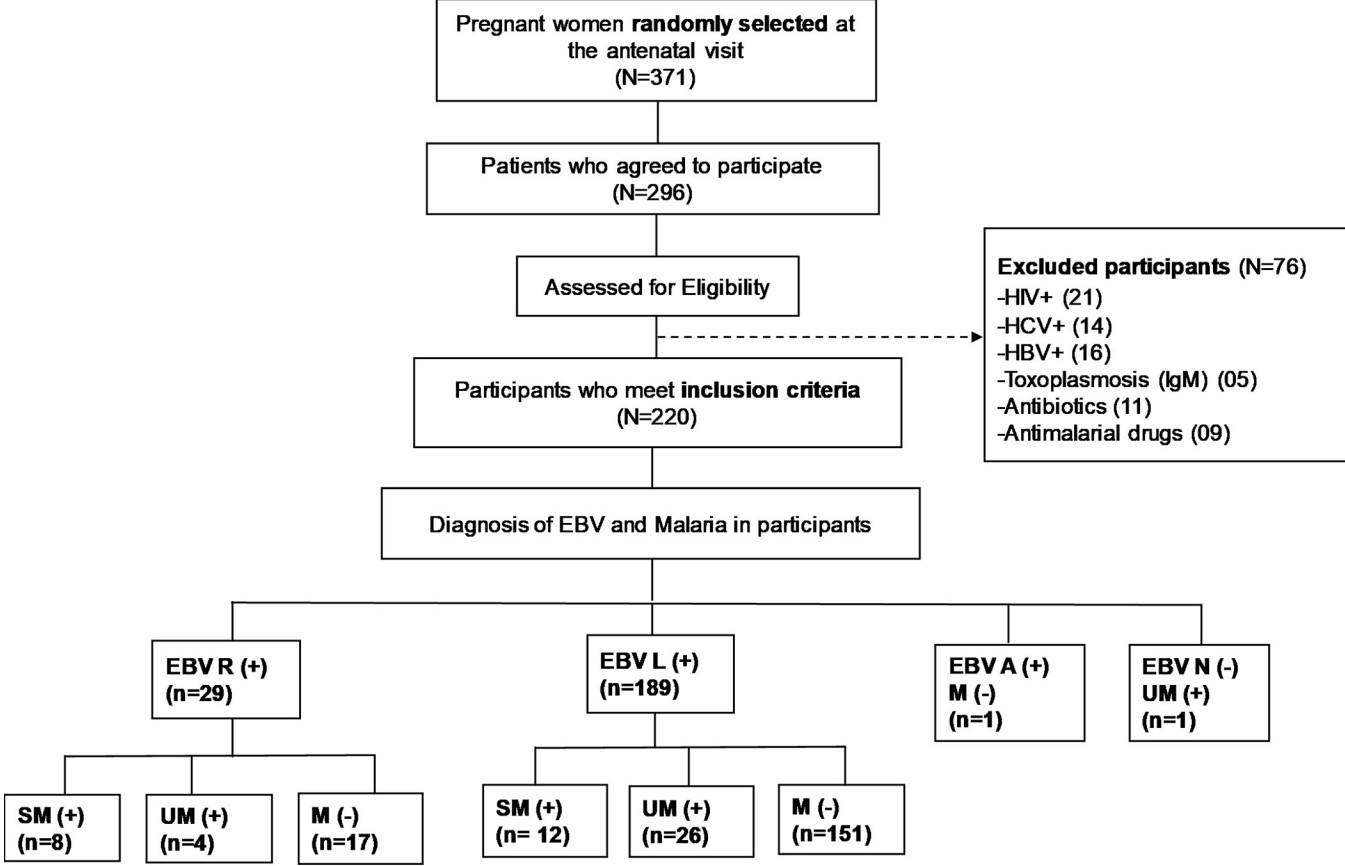

**Fig 2. Flow chart of the study describing the strategy for recruiting participants in 03 hospitals in the West region of Cameroon.** Ethical clearance and administrative authorizations were obtained from the relevant authorities. Based on EBV and malaria infection status, study participants were classified into 8 groups (EBV: Epstein-Barr virus, M: malaria, R: reactivation, UM: uncomplicated malaria, SM: severe malaria, L: latent, N: Negative and A: Acute, (+): presence, (-): absence).

0.55 respectively). Furthermore, severe anemia was strongly associated with malaria during pregnancy (AOR 8.35, 95% CI 1.68–50.06). Malaria cases were more frequent in primiparous women than multiparous women (62.7% vs. 37.3%). However, maternal age, study site, and gestational age were not statistically associated with malaria.

### *Plasmodium* species identification by Nested PCR

The 18S rRNA genes specific to *Plasmodium* species were amplified by Nested PCR. Three *Plasmodium* species were identified among the 51 pregnant women, namely *P. falciparum*, *P. malariae*, and *P. ovale*. *P. falciparum* was the most frequent species (23.2%, 51/220), followed by *P. malariae* (3.6%, 8/220) and *P. ovale* (0.9%, 2/220). *P. falciparum* mono-infections were more common (18.6%, 41/220) than mixed infections (4.5%, 10/220) (Fig 3). No non-*falciparum* mono-infections were reported in this study.

### Parasitemia in pregnant women infected with *Plasmodium*

Based on clinical complications and parasitemia results, twenty pregnant women had severe malaria, and thirty-one pregnant women had uncomplicated malaria. The level of parasitemia ranged from 764 to 70020 parasite/μl. The median parasitemia was 4057 parasites/μl (IQR: 1110–25371 parasites/μl). 19.6% (10/51) had a parasitemia <1000 parasites/μl, 41.2% (21/51)

**Table 2. Socio-demographic characteristics of study participants from three health centers in the West region of Cameroon, January to September 2022.**

| Variables | Frequency* (n = 220) |
|---|---|
| **Median age in years** | 26 years, IQR: 23–31 |
| **Marital status** | |
| Single | 59 (26.8%) |
| Married | 90 (40.9%) |
| In relationship | 71 (32.3%) |
| **Education** | |
| Primary | 8 (3.6%) |
| Secondary | 151 (68.6%) |
| High Education | 57 (25.9%) |
| Illiterate | 4 (1.8%) |
| **Environment** | |
| Rural | 74 (33.6%) |
| Urban | 146 (66.4%) |
| **Occupation** | |
| Student | 47 (21.4%) |
| Employed | 63 (28.6%) |
| Farmer | 41 (18.6%) |
| Unemployed | 69 (31.4%) |
| **Maternal income** | |
| Low | 77 (35%) |
| Middle | 143 (65%) |
| **Parity** | |
| Primigravidae | 79 (35.90%) |
| Multigravidae | 141 (64.1%) |
| **Gestational age** (21 Weeks, IQR: 16–27) | |
| 1$^{rst}$ trimester (<14 weeks) | 44 (20%) |
| 2$^{nd}$ trimester (14–27 weeks) | 123 (55.9%) |
| 3$^{rd}$ trimester (≥ 28 weeks) | 53 (24.1%) |
| **Ethnic group** | |
| Bamileke | 204 (92.7%) |
| Bororo | 5 (2.3%) |
| Foulbe | 3 (1.4%) |
| Others † | 8 (3.6%) |

\* Categorical data were summarised by frequencies/percentages and continuous data such as age and gestational age by the median (IQR)

† People from the North West and South West regions of Cameroon.

had a parasitemia between 1000–9999 parasites/μl, and 39.2% (20/51) had a parasitemia ≥10000 parasites/μl. The median parasitemia among pregnant women with severe malaria was 32742 parasites/μl (IQR: 23196–44393 parasites/μl) and 2062 parasites/μl (IQR: 933–3161 parasites/μl) among pregnant women with uncomplicated malaria. Pregnant women with mixed *P. falciparum* and *P. malariae* infections had significantly higher median parasitemia than pregnant women with *P. falciparum* mono-infections (31604 parasites/μl, IQR: 16899–49572 parasites/μl vs. 2578 parasites/μl, IQR: 1028–15889 parasites/μl; p = 0.0007) (Fig 4). However, most severe malaria cases were due to *P. falciparum* mono-infections (60%), while 40% were due to mixed infections.

**Table 3. Multivariable analysis of associated factors for malaria among pregnant women in the West region of Cameroon.**

| Variables | Malaria Status | | Crude Analysis* | | | Adjusted Analysis† | | |
|---|---|---|---|---|---|---|---|---|
| | Negative (n = 169) | Positive (n = 51) | OR | 95% CI | p-value | OR | 95% CI | p-value |
| **Environment** | | | | | | | | |
| Rural | 39 (23.1%) | 35 (68.6%) | - | | | - | | - |
| Urban | 130 (76.9%) | 16 (31.4%) | 0.14 | 0.07–0.27 | <0.001 | 0.12 | 0.05–0.28 | <0.001 |
| **Use of LLINs*** | | | | | | | | |
| No | 54 (32%) | 33 (64.7%) | - | | | - | | - |
| Yes | 115 (68%) | 18 (35.3%) | 0.26 | 0.13–0.49 | <0.001 | 0.35 | 0.14–0.84 | 0.021 |
| **IPT** | | | | | | | | |
| No | 48 (28.4%) | 31 (60.8%) | - | | | - | | - |
| Yes | 121 (71.6%) | 20 (39.2%) | 0.26 | 0.13–0.49 | <0.001 | 0.23 | 0.09–0.55 | 0.001 |
| **Severe anemia**(<7g/dl)** | | | | | | | | |
| No | 165 (97.6%) | 40 (78.4%) | - | | | - | | - |
| Yes | 4 (2.4%) | 11 (21.6%) | 11.34 | 3.67–42.65 | <0.001 | 8.35 | 1.68–50.06 | 0.013 |
| **No anemia (≥11g/dl)** | | | | | | | | |
| No | 60 (35.5%) | 34 (66.7%) | - | | | - | | - |
| Yes | 109 (64.5%) | 17 (33.3%) | 0.28 | 0.14–0.53 | <0.001 | 0.34 | 0.13–0.83 | 0.019 |
| **Parity** | | | | | | | | |
| Multigravidae | 122 (72.2%) | 19 (37.3%) | - | | | - | | - |
| Primigravidae | 47 (27.8%) | 32 (62.7%) | 4.37 | 2.28–8.59 | <0.001 | 6.14 | 2.55–15.91 | <0.001 |

\* Estimated by univariate logistic regression

† Estimated by multivariate logistic regression including all above variables

\*Long-lasting insecticidal nets.

\*\* The anemia cut-offs for pregnant women as defined by the World Health Organization [53]

We compared malaria symptoms in patients with *P. falciparum* mono-infections and those with mixed infections (S1 Fig). We found that anemia (70.6%), fever (71.4%), fatigue (75.6%), sore muscle (78.8%), nausea (85.7%), headache (74.3%), and respiratory distress (80%) were more frequent in patients with *P. falciparum* mono-infections than those with mixed infections (p = 0.001).

## Prevalence of Epstein-Barr virus reactivation in pregnant women

The overall prevalence of EBV in pregnant women was 99.5% (219/220). Thirteen-point two percent (13.2%, 29/220) had EBV reactivation, of whom 5.5% (12/220) were coinfected with malaria, and 7.7% (17/220) had no malaria. Eighty-five-point nine percent (85.9%, 189/220) had a latent infection, 0.5% (1/220) had an acute infection, and 0.5% (1/220) had no previous infection (Fig 5A). The serological pattern of EBV infection is presented in Fig 5B below.

No association was found between stress, anxiety, alcohol consumption, age, parity, and gestational age with EBV reactivation. Symptoms of EBV reactivation appeared to be statistically similar to those of latent infection. No significant differences were found between these groups.

## Association between EBV reactivation and malaria severity in pregnant women

Fig 6 shows the distribution of malaria in the two EBV groups, one latent and the other reactivated. A total of 29 pregnant women (29/220) had reactivated EBV, and 189 (189/220) had

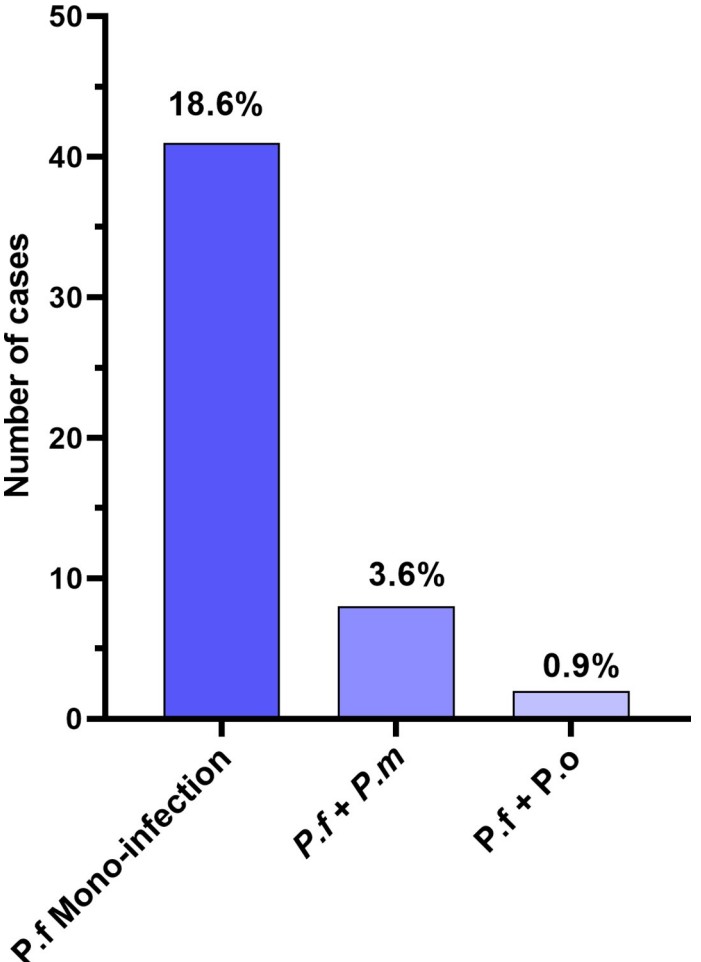

**Fig 3. Frequency of *Plasmodium* species, mono-infection, and mixed infection among infected women.** *P.f:* *Plasmodium falciparum*, *P.m*: *Plasmodium malariae*, and *P.o*: *Plasmodium ovale*.

latent EBV. Among the group with reactivated EBV, 12 (12/29) had malaria, including 8 (8/12) with severe malaria and 4 (4/12) with uncomplicated malaria. In the latent EBV group, 38 had malaria, of whom 12 (12/38) were severe malaria and 26 (26/38) uncomplicated cases. A high proportion of severe malaria was observed in pregnant women with reactivated EBV (66.7% vs. 31.6%), while most pregnant women with latent EBV were more affected by uncomplicated malaria (68.4% vs. 33.3%). Pregnant women with reactivated EBV were more likely to develop severe malaria than women with latent EBV (OR 4.33, 95% CI 1.08–17.25, p = 0.03).

## The Link between parasitemia and EBV status

The level of parasitemia in patients with malaria was plotted against EBV infection status, as shown in Fig 7. We found that pregnant women with latent EBV had lower levels of parasitemia than pregnant women with EBV reactivation (median, 2816 vs. 19002 parasites/µl; p = 0.02). EBV reactivation was not associated with malaria symptoms.

We compared the parasitemia of malaria patients infected with different *Plasmodium* species with the levels of IgM VCA, IgG EBNA, and IgG EA (D) antibodies. No significant association was found between parasitemia and the level of EBV antibodies (p = 0.57, p = 0.79, p = 0.17, respectively) (S2 Fig).

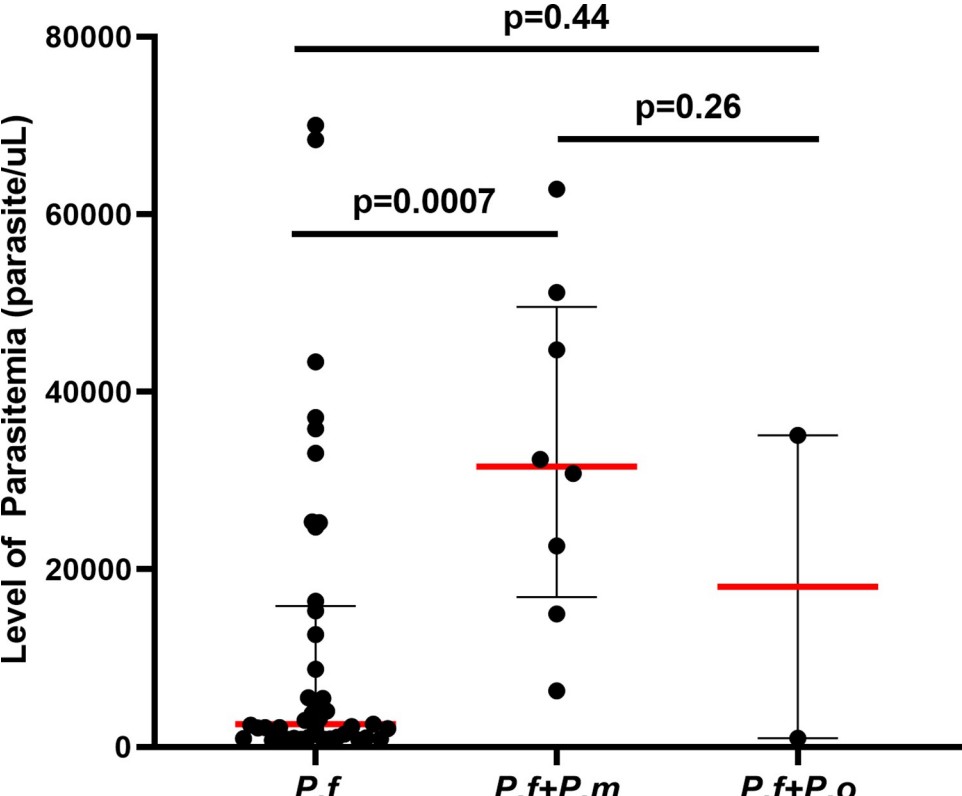

**Fig 4. Level of parasitemia in malaria patients with *P. f* mono-infection (n = 41), mixed *P. falciparum* and *P. malariae* infection (n = 8); and mixed *P. falciparum* and *P. ovale* infection (n = 2).** Individual data points represent the parasite density. The scatter dot plots with lines show the median with interquartile range (IQR). The Mann-Whitney U test was used for statistical analysis.

Finally, we assessed the EBV antibody responses in participants with severe malaria, uncomplicated, and without malaria. The level of EBV antibody ranged from 0.8 to 114.4 U/ml for IgM VCA, from 16.9 to 213.1 U/ml for IgG EBNA, and from 0.05 to 3.6 (IP) for IgG EA (Fig 8). The response of VCA IgM was significantly higher in patients with severe malaria compared to patients with uncomplicated malaria or without malaria (median, 10.32 U/ml, p = 0.006). IgG EBNA was high in all groups, but increased concentrations were reported in patients who did not have malaria (median, 155.4U/ml, p<0.0001). The response of IgG EA was comparable in all groups; no significant differences were found (p = 0.54).

## Discussion

Despite compelling evidence that acute Epstein-Barr virus (EBV) infection negatively affects the development of humoral immunity during *Plasmodium* infection [27–29], the impact of EBV reactivation on malaria severity is not elucidated. The present study aimed to investigate the association between EBV reactivation and malaria severity in women living in a malaria-endemic region of Cameroon and to estimate the burden of EBV reactivation during malaria infection. The major finding is that pregnant women with EBV reactivation were more likely to develop severe malaria than those with latent EBV. This result was supported with lower median parasitemia in pregnant women with latent EBV compared to pregnant women with EBV reactivation. Our study is the first of its kind that revealed that pregnant women with a reactivated EBV had a higher chance of manifesting severe malaria than pregnant women with

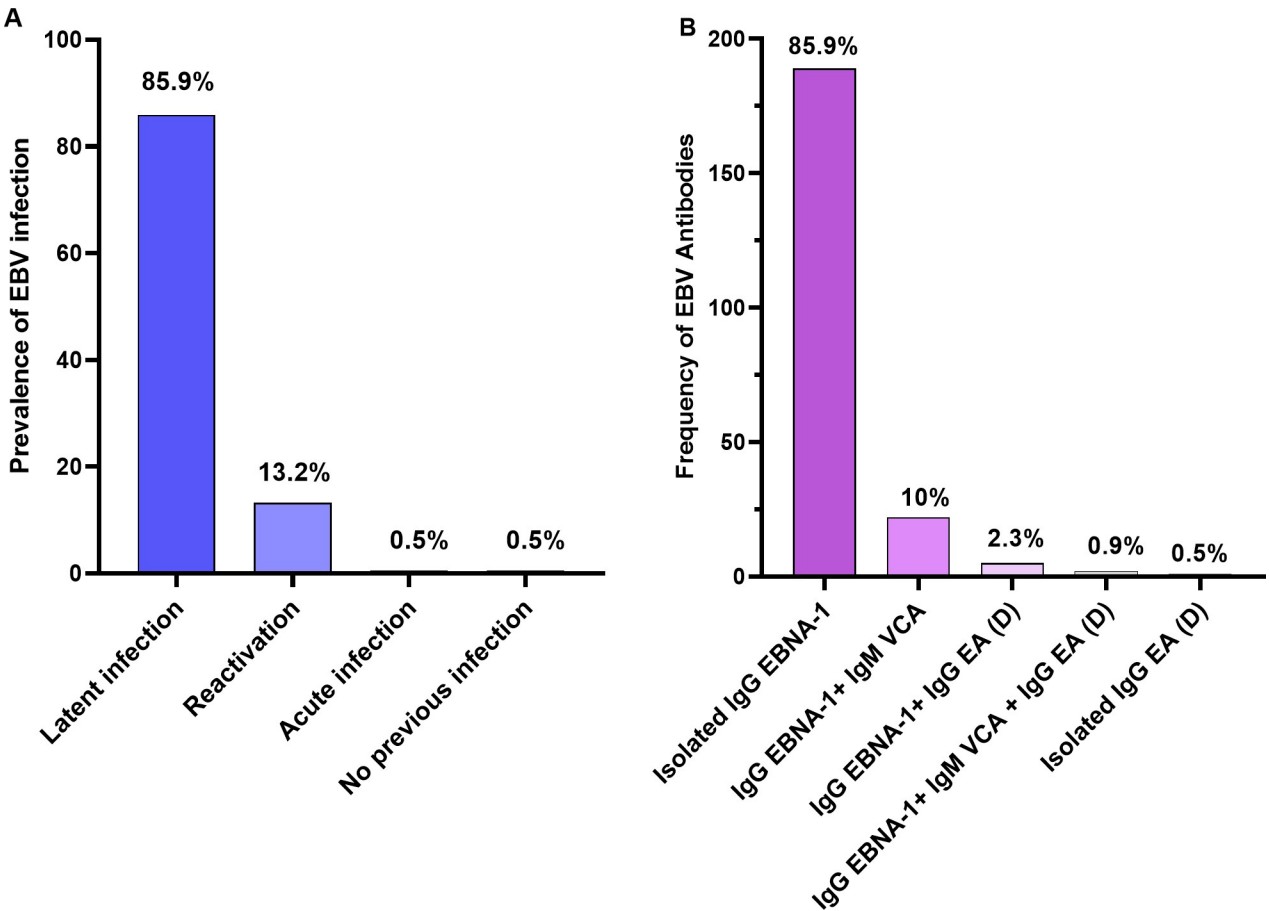

**Fig 5.** Prevalence of different stages of EBV infection in all participants (A) and Frequency of EBV serological pattern observed in the study population (B).

a latent EBV. This may be explained by the ability of EBV to disrupt the functional capacity of immune cells that it infects when it reactivates, namely monocytes, macrophages, dendritic Cells (DCs), NK cells, and neutrophils [54–57]. These cells are essential in controlling *P. falciparum* parasitemia thanks to their phagocytic activity, cytokine production, and antigen presentation [58, 59]. Therefore, a dysregulation of these functions could negatively affect the anti-malarial immune response, leading to severe forms of the disease. A recent study conducted by Hem Chandra Jha's team also supports that lytic reactivation of EBV infection during malaria may facilitate the development of cerebral malaria [30]. Here, we sought to investigate whether the lytic reactivation of EBV infection could influence the parasitemia level in pregnant women infected with *Plasmodium*. We found that pregnant women with latent EBV had lower levels of parasitemia than pregnant women with EBV reactivation (median, 2816 vs. 19002 parasites/μl). This suggests that EBV reactivation may impair the control of the parasitemia in individuals infected with *Plasmodium*. This finding is in accordance with a previous study that reported high and prolonged parasitemia in marmoset mice co-infected with *P. brasilianum* and EBV [60]. It has also been documented that acute EBV infection suppresses anti-malarial humoral responses in C57BL/6 mice infected with *Plasmodium yoelii XNL* and causes a defect in antibody production, leading to a loss of control of peripheral parasitemia [27, 28]. In this context, it is possible to speculate that during EBV reactivation, the virus induces high

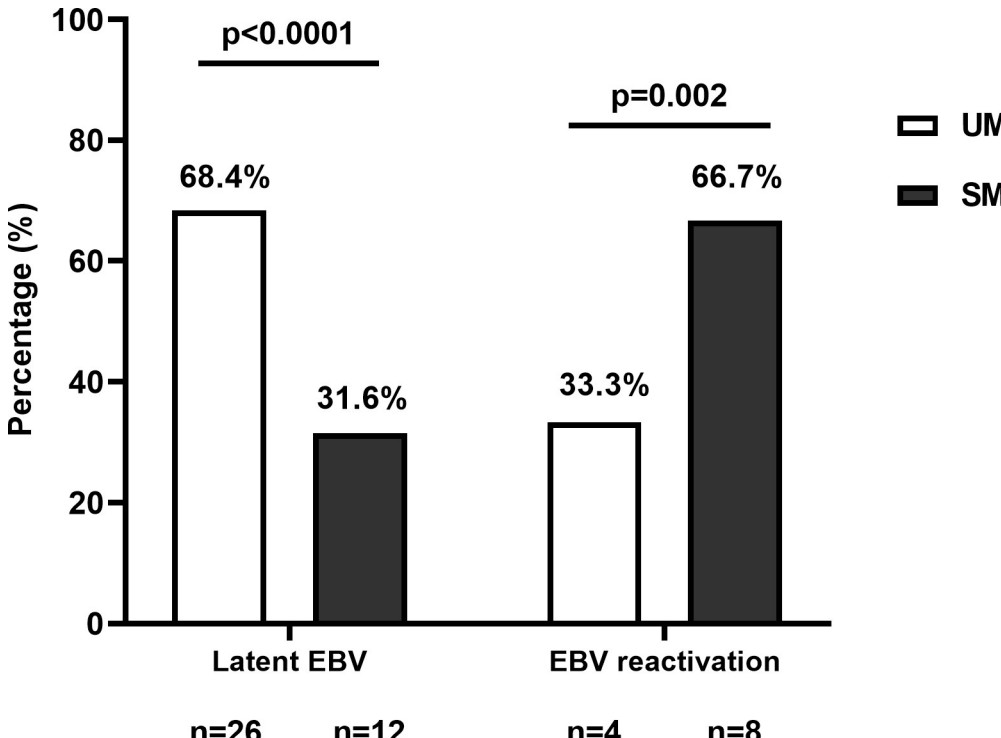

**Fig 6. Distribution of malaria among pregnant women coinfected with a latent EBV and a reactivated EBV.** SM: Severe malaria; UM: uncomplicated malaria; M (-): Malaria Negative. The Mann-Whitney U test was used for statistical analysis.

secretion of cytokines such as TNF-α, IL-12p40, IL-10, and IFN-γ, which could exacerbate the immune response to malaria and contribute to uncontrolled proliferation of the parasite. Nevertheless, it is also possible that EBV reactivation could serve as a hallmark of malaria severity during pregnancy or a hallmark of an unfavorable pregnancy outcome.

The overall prevalence of symptomatic malaria at the enrolment was 23.2%. This prevalence was similar to the prevalence reported in Bafang (25%) [61] and Dschang (25.3%) [62] in West region of Cameroon, suggesting that the occurrence of malaria infection in this region is moderate and can be considered mesoendemic [63]. Indeed, malaria transmission in this region has been previously described as relatively stable, with entomological inoculation rates Dvarying from 62.8 to 90.5 infectious bites/per person/year [36, 37]. It is important to mention that the study was carried out during the dry and rainy seasons when transmission of *Plasmodium* is high to better estimate malaria prevalence. Furthermore, 64.1% of pregnant women had taken IPT, and 60.5% were using LLINs at the time of the study, which may have also contributed to the decrease in the burden of malaria. Primiparous pregnant women living in rural areas were more susceptible to *Plasmodium* infection than multiparous women. This result is consistent with previous studies showing that malaria is associated with parity and residence in rural areas [64–66]. However, no statistically significant association was found between malaria status in pregnant women according to their age, study site, and gestational age (p>0. 05), although some studies found that young women (<25 years) were more at risk of malaria than older women and women in the first and second trimester were more vulnerable to malaria [67–70].

*P. falciparum* was the most common, followed by *P. malariae* and *P. ovale. P. falciparum* mono-infections were more common than mixed infections. About 60% of severe malaria

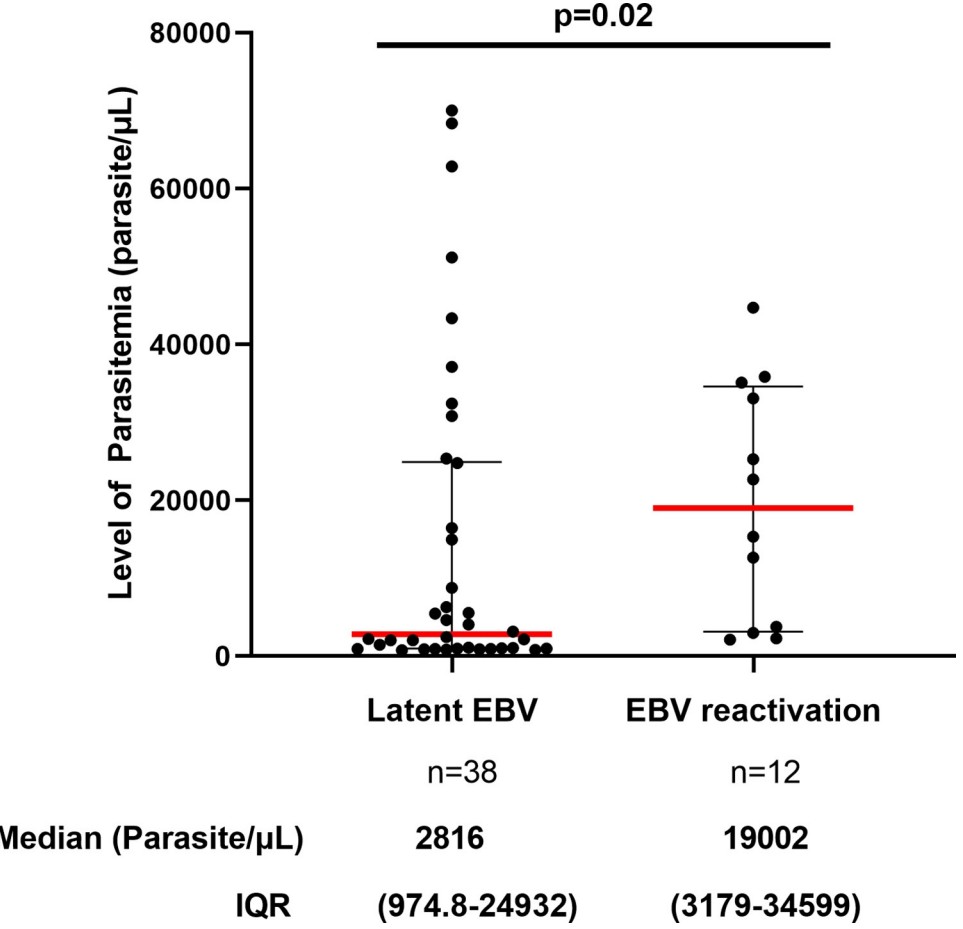

**Fig 7. Comparison of parasitemia level in malaria patients with latent infection and EBV reactivation.** Individual data points represent the parasite density. The scatter dot plots with lines show the median with interquartile range (IQR). Data at the bottom represent the median (parasite/μl) and IQR values of each group. The Mann-Whitney U test was used for statistical analysis (p = 0.02).

cases were due to *P. falciparum* mono-infections. This result is not surprising as it has been reported that *P. falciparum* is the most widespread and pathogenic malaria parasite associated with severe forms of illness, particularly in the WHO African region [71, 72]. However, our study underlines the urgency of strengthening measures to control the transmission of *P. falciparum* malaria and the need to develop effective therapeutic strategies. Indeed, previous studies revealed that most *P. ovale* and *P. malariae* infections are associated with *P. falciparum* infections [73–75].

The median parasitemia in infected pregnant women was slightly lower than the median parasitemia reported in Colombian pregnant women (4057 vs. 4400 parasites/μl) [76]. This is probably due to the use of intermittent preventive treatment and early malaria diagnosis. Women with mixed *P. falciparum* and *P. malariae* infections had significantly higher parasitemia than women with *P. falciparum* mono-infections and women with mixed *P. falciparum* and *P. ovale* infections (p = 0.01). This result may be explained by the predominance of *P. falciparum* parasite density in mixed infections. Indeed, *P. falciparum* is the only *Plasmodium* species that is responsible for high levels of parasitemia due to its ability to infect all red blood cells [77, 78], unlike *P. ovale*, which only infects young erythrocytes [79, 80] and *P. malariae*, older erythrocytes [81, 82] leading to a generally lower parasite density [73, 83]. No significant

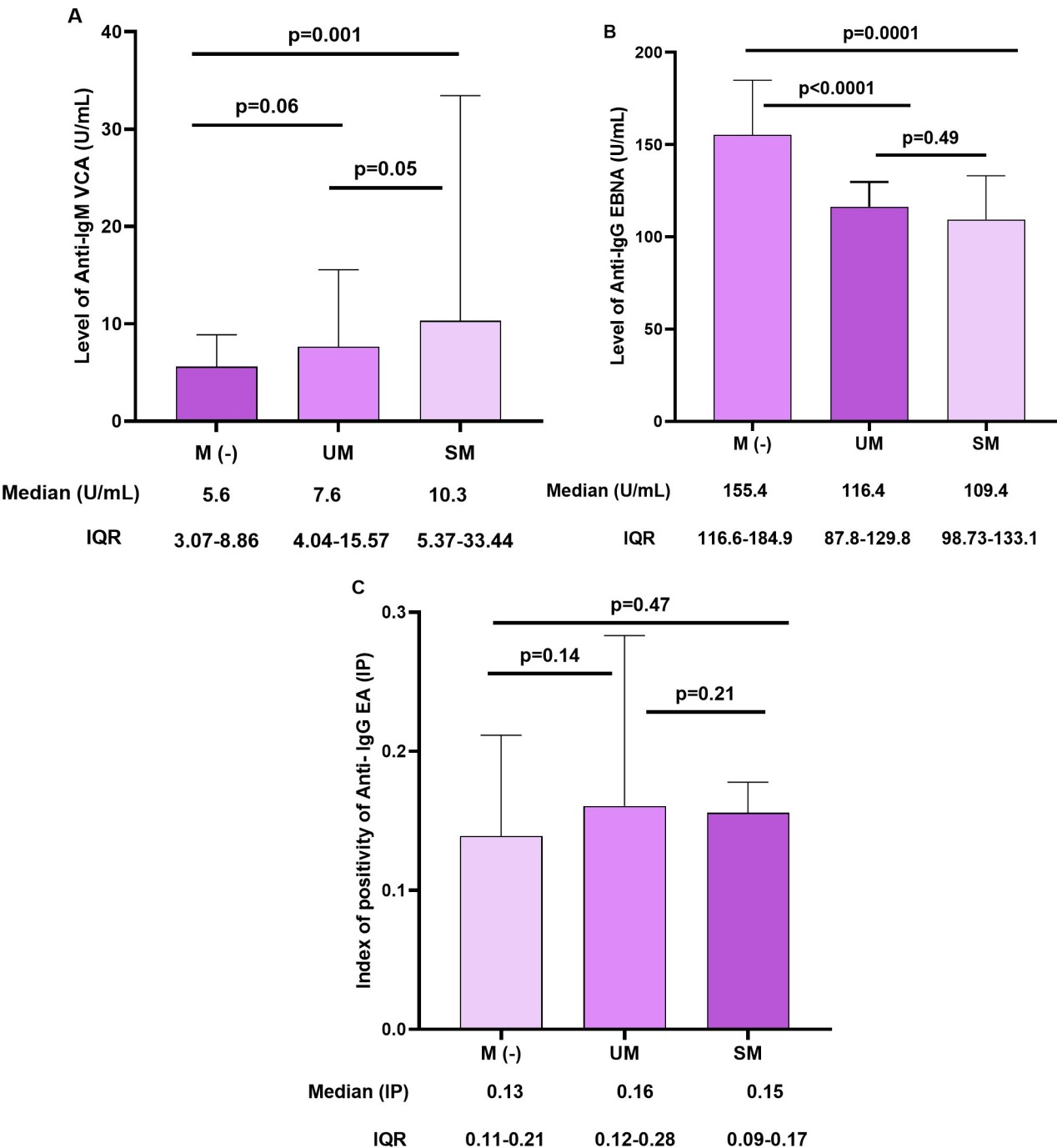

**Fig 8. Comparison of EBV-specific antibodies responses in patients with severe malaria (SM), uncomplicated malaria (UM), and patients without malaria (M (-).** For each graph, the column bars represent the median of the different EBV antibody concentrations (A and B) and the median of the positivity index (C), respectively. The data at the bottom represent median and IQR values. Statistical analysis was performed by the Mann-Whitney U test.

difference was found between malaria symptoms in patients with *P. falciparum* mono-infection and those with mixed infection (p = 0.88).

Almost all participants (99.5%) had been in contact with EBV in their lifetime, which is in line with previous studies that have shown a high rate of EBV seroprevalence in pregnant

women in Africa (100%) [31]. The prevalence of EBV reactivation in pregnancy was relatively low in our study (13.2%) compared to the study conducted by Haeri S. *et al.*, which showed a 35% EBV reactivation rate during pregnancy [20]. This could be due to the difference in methods used to diagnose EBV reactivation, the heterogeneity of the study population, and differences in the inclusion criteria. Indeed, in our study, ELISA method was used to detect IgM VCA, IgG EA, and IgG EBNA, three important markers in serological diagnosis of EBV infection. However, the detection of IgG VCA would also have been useful to have serological patterns that could indicate reactivation [52, 84]. We observed that only 5.5% of pregnant women were coinfected with malaria and had a reactivated EBV. This is inconsistent with a previous study showing elevated EBV DNA loads in pregnant women with malaria [31]. It is plausible that EBV reactivation in individuals with malaria depends on other factors, such as the severity of malaria, the number of malaria attacks, and the intensity of malaria transmission in the area [85].

Our study found no significant correlation between parasitemia and the level of EBV antibodies. In addition, EBV reactivation was not associated with malaria symptoms (p>0.05). A similar result was reported by Budiningsih *et al.* in Indonesia [86]. However, we observed that patients with severe malaria had significantly higher IgM VCA response compared to patients with uncomplicated malaria or without malaria, but IgG EBNA was relatively high in patients who did not have malaria. High levels of IgM VCA usually indicate primary infection but might also reappear in EBV reactivation [51]. In our study, the simultaneous presence of IgM VCA and IgG EBNA in patients with severe malaria reflects EBV reactivation. On the other hand, the isolated presence of EBNA IgG in patients without malaria would indicate the virus latency. This is consistent with some previous studies that have shown high EBV-specific antibody levels in children and pregnant women with acute malaria in hyperendemic regions [31, 87].

The first limitations of this study were the small sample of EBV reactivation/malaria (12/ 220), the inability to follow up with women with severe malaria and EBV reactivation to assess the outcome of malaria, the efficacy of antimalarial treatment, and their ability to recover during and after treatment. Secondly, the prevalence of reactivation during pregnancy may have been underestimated due to the lower sensitivity of the ELISA compared to the quantitative PCR. It is also interesting to highlight that, the detection of EBV antibodies by ELISA has limitations, such as the lack of specificity, the difficulty in staging EBV infection, and the equivocal interpretation of some antibody patterns [88]. The measurement of EBV DNA load should be used complementarily with antibody detection to diagnose EBV reactivation (viral replication) in patients.

## Conclusion

This study conducted in the Mifi and Nde divisions, a malaria endemic area in Cameroon showed that pregnant women who had EBV reactivation were more likely to develop severe malaria. This result was supported with lower median parasitemia in pregnant women with latent EBV compared to those with EBV reactivation. The response of VCA IgM was higher in patients with SM than those with uncomplicated malaria or without malaria, while IgG EBNA increased in patients without malaria. This suggests that EBV reactivation should be considered a risk factor for severe malaria in pregnant women living in malaria-endemic regions. Further studies are needed to understand how EBV reactivation disrupts the control of parasitemia and the anti-malarial immune response. It will also be interesting to assess the effect of EBV reactivation on placental malaria and investigate whether EBV reactivation increases the sequestration of *P. falciparum* parasites in the placenta.

## Supporting information

**S1 Fig. Frequency of malaria symptoms among pregnant women with a *P. falciparum* mono-infection (n = 41) and mixed infection (n = 10).** The Mann-Whitney U test was used for statistical analysis.
(TIF)

**S2 Fig. Correlation between the parasitemia in malaria patients infected with *P. falciparum* mono-infection (n = 41) and mixed infection (n = 10) with EBV antibodies responses (IgM VCA, IgG EA, and IgG EBNA.** Statistical analysis was done by linear regression ($R^2$).
(TIF)

**S1 Appendix. Questionnaire.**
(PDF)

## Acknowledgments

We are very grateful to Dr. Pawan Malhotra, a recipient of JC Bose Fellowship awarded by SERB, govt Of India, for hosting and supervising this work in the Malaria Biology Laboratory at the ICGEB. We also extend our gratitude to Dr. Emmanuel Haddison, Director of the Bangangte District Hospital, Dr. Ambassa Elime Gregoire, Director of the Mifi District Hospital of Bafoussam, Dr. Tchoukoua Serge, Assistant General Administrator of Universite des Montagnes Teaching Hospital, Dr. Yawat Djogang Anselme Michel, Prof. Pierre Rene Fotsing Kwetche, Dr. Simo Louokdom Josue and all the pregnant women who voluntarily agreed to take part in this study. We also express our sincere appreciation to all the nurses, midwives, and laboratory technicians who assisted us during registration, sample collection, and microscopic analysis and to all the members of the Malaria Biology Laboratory at the International Centre for Genetic Engineering and Biotechnology in New Delhi for their constructrive suggestions and overall support during this research work.

## Author Contributions

**Conceptualization:** Ide Armelle Djuidje Chatue, Severin Donald Kamdem, Palmer Masumbe Netongo.

**Data curation:** Ide Armelle Djuidje Chatue.

**Formal analysis:** Ide Armelle Djuidje Chatue, Severin Donald Kamdem.

**Funding acquisition:** Ide Armelle Djuidje Chatue, Pawan Malhotra, Palmer Masumbe Netongo.

**Investigation:** Ide Armelle Djuidje Chatue, Iqbal Taliy Junaid.

**Methodology:** Ide Armelle Djuidje Chatue, Severin Donald Kamdem, Franklin Maloba, Palmer Masumbe Netongo.

**Project administration:** Maximilienne Ascension Nyegue, Palmer Masumbe Netongo.

**Resources:** Pawan Malhotra, Palmer Masumbe Netongo.

**Supervision:** Maximilienne Ascension Nyegue, Pawan Malhotra, Palmer Masumbe Netongo.

**Validation:** Ide Armelle Djuidje Chatue, Severin Donald Kamdem, Iqbal Taliy Junaid, Pawan Malhotra, Palmer Masumbe Netongo.

**Visualization:** Ide Armelle Djuidje Chatue.

**Writing – original draft:** Ide Armelle Djuidje Chatue.

**Writing – review & editing:** Ide Armelle Djuidje Chatue, Maximilienne Ascension Nyegue, Severin Donald Kamdem, Franklin Maloba, Iqbal Taliy Junaid, Pawan Malhotra, Palmer Masumbe Netongo.

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
