## [Decision Letter · Decision Letter 0]

20 Dec 2023

PGPH-D-23-01980

Association between Epstein-Barr virus reactivation and severe malaria in pregnant women living in a Malaria-endemic region of Cameroon

Dear Dr. Djuidje Chatue,

Thank you for submitting your manuscript to PLOS Global Public Health. After careful consideration, we feel that it has merit but does not fully meet PLOS Global Public Health’s publication criteria as it currently stands. Therefore, we invite you to submit a revised version of the manuscript that addresses the points raised during the review process.

We look forward to receiving your revised manuscript.

Kind regards,

Mathieu Nacher

Academic Editor

Journal Requirements:

1. Please update your online Competing Interests statement. If you have no competing interests to declare, please state: “The authors have declared that no competing interests exist.”

2. Please provide separate figure files in .tif or .eps format only and remove any figures embedded in your manuscript file. Please also ensure that all files are under our size limit of 10MB. You may leave the figure captions or legends in the manuscript.

3. We notice that your supplementary figures are included in the manuscript file. Please remove them and upload them with the file type 'Supporting Information'. Please ensure that each Supporting Information file has a legend listed in the manuscript after the references list.

4. We do not publish any copyright or trademark symbols that usually accompany proprietary names, eg (R), (C), or TM  (e.g. next to drug or reagent names). Please remove all instances of trademark/copyright symbols throughout the text, including ® on pages 9, 10, 11 and 12.

5. Some material included in your submission may be copyrighted. According to PLOS’s copyright policy, authors who use figures or other material (e.g., graphics, clipart, maps) from another author or copyright holder must demonstrate or obtain permission to publish this material under the Creative Commons Attribution 4.0 International (CC BY 4.0) License used by PLOS journals. Please closely review the details of PLOS’s copyright requirements here: PLOS Licenses and Copyright. If you need to request permissions from a copyright holder, you may use PLOS's Copyright Content Permission form.

Potential Copyright Issues:

Figure 1: please (a) provide a direct link to the base layer of the map (i.e., the country or region border shape) and ensure this is also included in the figure legend; and (b) provide a link to the terms of use / license information for the base layer image or shapefile. We cannot publish proprietary or copyrighted maps (e.g. Google Maps, Mapquest) and the terms of use for your map base layer must be compatible with our CC-BY 4.0 license. 

Additional Editor Comments (if provided):

Reviewers' comments:

Reviewer's Responses to Questions

**Comments to the Author**

1. Does this manuscript meet PLOS Global Public Health’s publication criteria? Is the manuscript technically sound, and do the data support the conclusions? The manuscript must describe methodologically and ethically rigorous research with conclusions that are appropriately drawn based on the data presented.

Reviewer #1: Yes

Reviewer #2: Yes

2. Has the statistical analysis been performed appropriately and rigorously?

Reviewer #1: Yes

Reviewer #2: Yes

3. Have the authors made all data underlying the findings in their manuscript fully available (please refer to the Data Availability Statement at the start of the manuscript PDF file)?

Reviewer #1: No

Reviewer #2: Yes

4. Is the manuscript presented in an intelligible fashion and written in standard English?

Reviewer #1: Yes

Reviewer #2: Yes

5. Review Comments to the Author

Reviewer #1: This article raises an important question on which there is very little data, that of the relation between severity of malaria during pregnancy and reactivation of EBV.

The work is well conducted, original and well written.

I congratulate the authors for the quality of the study and of the paper, which will be very useful for the malarialogist community.

However, the article would benefit from lighter coverage of the secondary issues and stronger emphasis on the main question - the link between severity and EBV reactivation - which is given little attention in the results and discussion.

Thus, the number of patients with EBV reactivation and severe malaria is not clearly presented. The main result is presented in reverse, even in the abstract, presenting the link between EBV reactivation and uncomplicated malaria. It would be much clearer to present it the other way round: link between EBV reactivation and severe malaria, presenting the numbers, which are likely to be small.

In detail, here are my comments:

Major corrections

Abstract

Present the main result with regards to severe malaria.

Introduction

L93 : and children ?

L94 : “can” instead of “cause”

L96: “decrease” instead of suppression

L120-121: not clear: what do you mean with “naturally clear the parasite”? without treatment? Are there situation when a pregnant woman with malaria is not treated? Even asympto? Please clarify.

L124: what do you mean with “possible outcome”? please specify

Material and method

L138: low-cost: care are not free for pregnant women in Cameroun?

L154-155: please clarify what you mean with “participants approached the hospitals”: this refers to reason of consultation? Or is it an inclusion criteria? If yes modify. Did you included only women consulting for fever?

L157-162: combine non-inclusion and exclusion criteria to avoid repetition, or explain the difference (timing ? it the serological results are known before consent?)

Why did you excluded women with antibiotics?

How did you defined depression?

L166: “all” instead of “various”?

L170: how did you proceed to the random sampling? Please specify

L173: P” is the prevalence of EBV and malaria coinfection among pregnant women in Kenya, which is 64% [31].”: it seems very high, it is among women consulting for fever? If yes specify

L177: please provide the questionnaire as a supplementary material

How did you defined the bednet use? In your results it is presented yes/no: what was the question? The night before the inclusion? Usually? Everyday?

Idem for IPT: how is defined yes or no? according to the trimester of pregnancy? At least one dose? What if very early pregnancy?

How did you measure anxiety?

L192: In my opinion, all the details concerning microscopy and PCR are too detailed. They should be grouped together in a paragraph of no more than 10 lines, with references to the methods used and the detection threshold.

Table 1 should be deleted.

However, please explain why you did the PCR on filter paper when you have whole blood? Was it sent to another laboratory? It would have been easier to perform the PCR directly on the blood and increase sensitivity. Please explain why.

L254-268: idem too detailed in my opinion, to be summarized

Please add the definition of severe malaria++

Results

L287 : and CMV ? inclusion criteria

L288-293: redundant with flow chart, could be deleted

Fig2: remove “ethical clearance…” from the title

The last line is very interesting but should be presented as a tree

Table 3: please define “maternal income”

L312: you did not presented gestational age as a continuous variable, and you did not present BMI, please correct

Table : please correct the table footer calls

As mentioned in the method, the use of IPT and LLIN must be defined, either in the method or here in the form of a footnote.

L328: must be deleted

Fig3: in my opinion, should be deleted as it does not provide any useful information

Fig4: would be more interesting to describe these % in the study population and not % of plasmodium positive

L343: first sentence to be removed

L347 and above: the number at the beginning of a sentence must be in letters, otherwise it must be rephrased.

Parasitemia in pregnant women: could provide the median parasitemia among women with severe malaria vs uncomplicated malaria?

L361-364: the % presented concern which population?

L367: provide the numbers N++

L379: “Symptoms of EBV infection were similar in patients with EBV reactivation or latent infection.” What do you mean? Please clarify

“Association between EBV reactivation and malaria severity in pregnant women”: this is the main result of your study but the results are not clear. Please provide the n/N for each category (severe/non severe malaria, EBV reactivation y/n, what did you do with the woman who had no EBV?)

And please calculate the OR with “severe malaria” as the outcome: it would be much clearer and help the reader understand your main result. This should also be included in your abstract.

Discussion

Start the discussion with a brief summary of your results (this comes too late, L514).

And point out that this is a first investigation of this question (which you say L519 > to move).

Then present the limitations/biases: your sample of EBV reactivation/malaria is small (N not clearly presented, 12?).

But even with this limitation, you have important results; you just need to specify the bias so that the reader can interpret the relevance of your work.

L441: not relevant

L459-460: not relevant

L467 and above: I don't think it's relevant to this article. There's no link with the objective of the study, so it's difficult for the reader to see the point.

L472-477: idem: in your study they were all infected by Pf, at least; so not relevant

L501-503: to be placed in the limitation

L510-513: It's a complex question that is not related to your main objective. I suggest you delete it to simplify your text. Otherwise, it would deserve much more explanation.

Reviewer #2: The authors evaluated the proportion of cases of severe and non-severe malaria in pregnant women randomly selected in 3 centres in Cameroon according to the presence of EBV reactivation. They showed that women with EBV reactivation were more often affected by severe malaria. A link was also established between EBV reactivation profile and parasitaemia.

The diagnostic method seems robust and the work useful for confirming the hypothesis of a link between reactivation and severe malaria in pregnant women.

The study initially included 220 patients, but in the end only 51 had malaria and the number of patients with reactivation and malaria was only 12, which is a very small number for further analysis especially a multivariate analysis was not done here which is somechat problematic considering the confoundig factors of either having positive EBV serology or severe malaria. Besides only serological analyses of EBV were done and no EBV viral load was obtained which could have been informative to ascertain the EBV reactivation profile and avoid possible false positive serological reactivati

---

## [Decision Letter · Decision Letter 1]

20 May 2024

PGPH-D-23-01980R1

Association between Epstein-Barr virus reactivation and severe malaria in pregnant women living in a Malaria-endemic region of Cameroon

Dear Dr. Djuidje Chatue,

Thank you for submitting your manuscript to PLOS Global Public Health. After careful consideration, we feel that it has merit but does not fully meet PLOS Global Public Health’s publication criteria as it currently stands. Therefore, we invite you to submit a revised version of the manuscript that addresses the points raised during the review process.

Your manuscript has been evaluated by one new reviewer, and their comments are appended below.

The reviewer has requested clarification of how severe malaria was diagnosed in this study, as well as an explanation of the relatively low incidence of severe malaria. The reviewer has also identified a few grammatical corrections for you to address. Please ensure you address each of the reviewer's comments when revising your manuscript.

We look forward to receiving your revised manuscript.

Kind regards,

Hugh Cowley

Staff Editor

Journal Requirements:

2. We ask that a manuscript source file is provided at Revision. Please upload your manuscript file as a .doc, .docx, .rtf or .tex.

Additional Editor Comments (if provided):

Reviewers' comments:

Reviewer's Responses to Questions

**Comments to the Author**

1. If the authors have adequately addressed your comments raised in a previous round of review and you feel that this manuscript is now acceptable for publication, you may indicate that here to bypass the “Comments to the Author” section, enter your conflict of interest statement in the “Confidential to Editor” section, and submit your "Accept" recommendation.

Reviewer #3: All comments have been addressed

2. Does this manuscript meet PLOS Global Public Health’s publication criteria? Is the manuscript technically sound, and do the data support the conclusions? The manuscript must describe methodologically and ethically rigorous research with conclusions that are appropriately drawn based on the data presented.

Reviewer #3: Yes

3. Has the statistical analysis been performed appropriately and rigorously?

Reviewer #3: Yes

4. Have the authors made all data underlying the findings in their manuscript fully available (please refer to the Data Availability Statement at the start of the manuscript PDF file)?

Reviewer #3: Yes

5. Is the manuscript presented in an intelligible fashion and written in standard English?

Reviewer #3: Yes

6. Review Comments to the Author

Reviewer #3: The authors set out to determine whether there is an association between the reactivation of EBV and the occurrence of severe malaria in pregnant women residing in a meso-endemic malaria zone of Cameroon. To answer this research question they conducted a cross-sectional hospital-based study in three locations in the west region of Cameroon. Using a structured questionnaire they recruited 220 consenting pregnant women, from whom they collected blood for diagnosis of malaria by microscopy and PCR and for preparation of plasma for serology of EBV. Reactivation of EBV was done by measuring IgM specific antibodies whereas IgG antibodies detection signified exposure. As expected more than 95 % of the participants were exposed to EBV. Severe malaria was significantly more prevalent in the subgroup of participants with reactivated EBV as compared to those exposed to latent EBV infection.

Severe malaria is usually more prevalent in patients with mega-parasitaemia in a hyper-endemic area. The authors should explain how severe malaria was diagnosed and discuss the relatively low incidence of severe malaria in their study.

There were a few grammatical slips which could be corrected by careful proof reading eg:

line 145 : Write 'study sites.' instead of 'study site'.

p228:Give the key of (+), (-), (+/-) in a legend under Table 1.

line 269:There is no ethnic group in Cameroon called Bamenda. Do you mean people from the North West region?

line 365: Write "We assessed.." instead of "we assess..."

line 388 : Write, "First of its kind.." instead of "first of its.."

line 390: Do not capitalize 'dendritic'

line 423: Write "However..."instead of "On the other hand..."

In conclusion , I find that the study was well planned and well executed. Its significance lies in the fact that it was conducted in an endemic area not previously studied for this kind of association. The article should be accepted after proofreading and correcting the errors.

7. PLOS authors have the option to publish the peer review history of their article (what does this mean?). If published, this will include your full peer review and any attached files.

**Do you want your identity to be public for this peer review?** For information about this choice, including consent withdrawal, please see our Privacy Policy.

Reviewer #3: No

---

## [Decision Letter · Decision Letter 2]

12 Jul 2024

Association between Epstein-Barr virus reactivation and severe malaria in pregnant women living in a Malaria-endemic region of Cameroon

PGPH-D-23-01980R2

Dear Mrs. Djuidje Chatue,

We are pleased to inform you that your manuscript 'Association between Epstein-Barr virus reactivation and severe malaria in pregnant women living in a Malaria-endemic region of Cameroon' has been provisionally accepted for publication in PLOS Global Public Health.

Best regards,

Julia Robinson

Executive Editor

Reviewer Comments (if any, and for reference):

Reviewer's Responses to Questions

**Comments to the Author**

1. If the authors have adequately addressed your comments raised in a previous round of review and you feel that this manuscript is now acceptable for publication, you may indicate that here to bypass the “Comments to the Author” section, enter your conflict of interest statement in the “Confidential to Editor” section, and submit your "Accept" recommendation.

Reviewer #3: All comments have been addressed

2. Does this manuscript meet PLOS Global Public Health’s publication criteria? Is the manuscript technically sound, and do the data support the conclusions? The manuscript must describe methodologically and ethically rigorous research with conclusions that are appropriately drawn based on the data presented.

Reviewer #3: Yes

3. Has the statistical analysis been performed appropriately and rigorously?

Reviewer #3: Yes

4. Have the authors made all data underlying the findings in their manuscript fully available (please refer to the Data Availability Statement at the start of the manuscript PDF file)?

Reviewer #3: Yes

5. Is the manuscript presented in an intelligible fashion and written in standard English?

Reviewer #3: Yes

6. Review Comments to the Author

Reviewer #3: I have reviewed this article before and foubd it acceptable, subject to minor concerns which have been addresse.....

d satisfactorily. Tge article shoukd ve accepted.

7. PLOS authors have the option to publish the peer review history of their article (what does this mean?). If published, this will include your full peer review and any attached files.

**Do you want your identity to be public for this peer review?** For information about this choice, including consent withdrawal, please see our Privacy Policy.

Reviewer #3: No
